# Genomic-Analysis-Oriented Drug Repurposing in the Search for Novel Antidepressants

**DOI:** 10.3390/biomedicines10081947

**Published:** 2022-08-11

**Authors:** Mohammad Hendra Setia Lesmana, Nguyen Quoc Khanh Le, Wei-Che Chiu, Kuo-Hsuan Chung, Chih-Yang Wang, Lalu Muhammad Irham, Min-Huey Chung

**Affiliations:** 1School of Nursing, College of Nursing, Taipei Medical University, Taipei 11031, Taiwan; 2Professional Master Program in Artificial Intelligence in Medicine, College of Medicine, Taipei Medical University, Taipei 11031, Taiwan; 3Research Center for Artificial Intelligence in Medicine, Taipei Medical University, Taipei 11031, Taiwan; 4Translational Imaging Research Center, Taipei Medical University Hospital, Taipei 11031, Taiwan; 5Department of Psychiatry, Cathay General Hospital, Taipei 10630, Taiwan; 6School of Medicine, Fu Jen Catholic University, New Taipei City 242062, Taiwan; 7Department of Psychiatry, School of Medicine, College of Medicine, Taipei Medical University, Taipei 11031, Taiwan; 8Department of Psychiatry and Psychiatric Research Center, Taipei Medical University Hospital, Taipei Medical University, Taipei 11031, Taiwan; 9Ph.D. Program for Cancer Molecular Biology and Drug Discovery, College of Medical Science and Technology, Taipei Medical University and Academia Sinica, Taipei 11031, Taiwan; 10Graduate Institute of Cancer Biology and Drug Discovery, College of Medical Science and Technology, Taipei Medical University, Taipei 11031, Taiwan; 11Faculty of Pharmacy, University of Ahmad Dahlan, Yogyakarta 55164, Indonesia; 12Department of Nursing, Shuang Ho Hospital, Taipei Medical University, New Taipei City 23561, Taiwan

**Keywords:** depression, genomic analysis, drug repurposing, functional annotation, bioinformatics, genetic, genomic variants, interleukin 6 receptor, sarilumab, satralizumab

## Abstract

From inadequate prior antidepressants that targeted monoamine neurotransmitter systems emerged the discovery of alternative drugs for depression. For instance, drugs targeted interleukin 6 receptor (*IL6R)* in inflammatory system. Genomic analysis-based drug repurposing using single nucleotide polymorphism (SNP) inclined a promising method for several diseases. However, none of the diseases was depression. Thus, we aimed to identify drug repurposing candidates for depression treatment by adopting a genomic-analysis-based approach. The 5885 SNPs obtained from the machine learning approach were annotated using HaploReg v4.1. Five sets of functional annotations were applied to determine the depression risk genes. The STRING database was used to expand the target genes and identify drug candidates from the DrugBank database. We validated the findings using the ClinicalTrial.gov and PubMed databases. Seven genes were observed to be strongly associated with depression (functional annotation score = 4). Interestingly, *IL6R* was auspicious as a target gene according to the validation outcome. We identified 20 drugs that were undergoing preclinical studies or clinical trials for depression. In addition, we identified sarilumab and satralizumab as drugs that exhibit strong potential for use in the treatment of depression. Our findings indicate that a genomic-analysis-based approach can facilitate the discovery of drugs that can be repurposed for treating depression.

## 1. Introduction

Depression is an emerging mental health problem affecting 322 million people around the world. Southeast Asia and the Western Pacific are the regions where depression is most prevalent [1]. Recent studies conducted in Taiwan reported that the prevalence of depression was 3.7–24.1% [2,3]. Some factors are classed as risk factors of depression, including genetic variation (single-nucleotide polymorphisms; SNPs) [4], female gender [5], chronic diseases [6], stressful life experiences [7], unemployment [8], low education level [8,9], poor physical exercise [10], lack of social support [11], and poor environment (high pollution) [12]. Furthermore, depression has been associated with poor quality of life, lack of well-being, disrupted daily activities, higher healthcare expenditure and utilization, and economic burden [13,14,15]. Thus, depression treatment remains one of the top public health priorities.

In terms of pathological neural substrates associated with depression, some brain regions have been discovered to be responsible for emotion and behavior [16]. The prefrontal cortex (PFC) is the highest level of the cerebral hierarchy, and is in charge of representing and carrying out actions [16]. The ventromedial cortex is a part of the PFC that has the function of controlling emotions resulting from autonomic and neuroendocrine system impulses [17]. A meta-analysis described that the roles of PFC, amygdala, and the hippocampus were crucial for depression development [18]. Increased amygdala activity was observed to be a precursor to depression [19]. In addition, prior studies have reported that the severity of depression is correlated with abnormal amygdala size and function [20,21]. Meanwhile, the hippocampus role has been excessively investigated due to its structure, which is rich in corticosteroid receptors, and its involvement in learning, memory, and neurogenesis processes [22,23]. A previous study described the association between the hippocampus and the hypothalamic–pituitary–adrenal (HPA) axis, in which negative impulses of stress lead to HPA axis activation by reducing neuronal plasticity in the hippocampus, resulting in a high level of cortisol, as one of the indicators of depression [24]. Moreover, according to Stawski et al., cortisol levels have been investigated in terms of their contribution to depression-related cognitive impairment [25]. Several studies also found that increasing cortisol secretion is associated with cognitive decline [26,27].

Barchas and Altemus proposed that the underlying pathomechanism for depression is related to the monoamine hypothesis [28]. This hypothesis postulates that the diminished availability of three key monoamine neurotransmitters (serotonin, norepinephrine, and dopamine) causes decreased neurotransmission and worsened cognitive function, both of which may contribute to depression [28]. Meanwhile, in the past decades, this hypothesis does not adequately explain the pathogenesis of depression [29]. Numerous studies point out the involvement of inflammatory processes in depression development [30,31,32]. The circulating of pro-inflammatory mediators in the brain is the consequence of the weakening of the semipermeable border of blood–brain barrier (BBB) due to chronical stress stimuli [33]. In depression, pro-inflammatory mediators such as IL6 have been linked to increasing the activity of the HPA axis by inducing the release of corticoliberin-releasing hormone (CRH) from the hypothalamus [34,35,36]. In term of serotonin synthesis, pro-inflammatory mediators (IL6, TNF, CRP) induced the activity of the HPA axis and an enzyme of tryptophan metabolism (indoleamine 2,3-dioxygenase), which led to lower levels of tryptophan and serotonin [36,37]. Prior studies have proposed that elevated IL6 and C-reactive protein (CRP) levels might predict the development of depressive symptoms [38,39,40]. Therefore, the exploration of the inflammatory hypothesis in depression raises the possibility of there being other biological processes that contribute to depression development, and opens the door to improving the treatment of depression.

The current treatment of depression is antidepressants, which was developed on the basis of monoamine neurotransmitter systems and target neural synapses [41]. Monoamine oxidase inhibitors (MAOIS) and tricyclic antidepressants are the first generation medications for depression; these produce serious adverse effects by blocking postsynaptic receptors [42]. For safety reasons, the second generation of antidepressants was developed, which includes selective serotonin reuptake inhibitors (SSRIs), selective noradrenaline reuptake inhibitors (SNRIs), dual serotonin and noradrenaline reuptake inhibitors, and multitarget antidepressants [42]. According to the National Institute for Clinical Excellence, SSRI medications are recommended to be the first-line treatment of depression [43]. However, SSRI medication often induces insufficient responses. Only 30% of individuals take commonly recommended antidepressant medications for depression remission [44], while 15–60% of depressed patients do not adequately response the medication [45]. Three in every ten patients with depression that are treated with antidepressants have reported treatment resistance [46]. Therefore, discovering alternative targets and potential medications for treating depression is urgent.

Drug repurposing is a common method for identifying potential new treatments using existing drugs [47,48]. The term “drug repurposing” refers to the repositioning of an existing medicine for a new indication [49]. For example, ketamine was originally approved by the United States Food and Drug Administration (USFDA) in 1970 for use as an intravenous anesthesia agent, but in 2019, it was approved for a new application: treatment-resistant depression [50,51]. Drug repurposing has some advantages over the conventional method of drug discovery; for example, drug repurposing candidates have already passed clinical trials for the original indication, and drug repurposing is faster and cheaper than the conventional method [7]. Furthermore, the mechanisms through which repurposed drugs affect the human body are usually already well established [47,52]. Therefore, the safety issues of repurposed drugs have been passed for the use in new medication.

Recent technological developments have encouraged researchers to consider common genetic variants, such as single-nucleotide polymorphisms (SNPs), in drug repurposing [53]. A popular method, established by Okada*,* et al. [54], involves utilizing a scoring system comprising eight functional annotations based on genomic analysis to prioritize target genes and discover drug repurposing candidates; the method was originally used to identify candidates for the treatment of rheumatoid arthritis according to SNP data collected from genome-wide association studies. Other studies have adapted Okada’s approach to use five sets of functional annotations to discover drug repurposing candidates for the treatment of atopic dermatitis [55] and asthma [56]. Functional annotations are considered crucial for evaluating diseases. Missense variants are nonsynonymous single-base changes that can cause changes in proteins [57]. Cis expression quantitative trait loci (*cis*-eQTL) are used to observed the variation in expressed genes in various tissues [58]. The Kyoto Encyclopedia of Genes and Genomes (KEGG) annotations are based on observing genetic associations that have an important role in molecular pathways [58]. Molecular pathway analysis related with protein–protein interactions (PPIs) consists of observing gene contributions to molecular functions in an organism [59]. Knockout mouse phenotype (KO mice) annotations exhibit considerable overlap with mammalian phenotype (MP) ontology annotations [60]. Accordingly, we postulate that a genomic-analysis-based approach using functional annotations could facilitate the discovery of candidates for drug repurposing for the treatment of depression.

Few studies have used SNP data to discover new drugs and drug repurposing candidates for the treatment of depression. A previous study involving the development of new drugs for treating major depressive disorder (MDD) focused only on genetic drug–target networks [61]. However, no study has adopted the genomic-analysis-based approach using functional annotations to identify drug repurposing candidates for the treatment of depression. In the present study, we prioritized potential target genes and drug repurposing candidates for depression by integrating SNP data from the Taiwan Biobank database with a machine learning algorithm by adopting a genomic-analysis-based approach and five sets of functional annotations (missense variant, *cis*-eQTL, KEGG, PPI, and KO mice).

## 2. Materials and Methods

### 2.1. Study Design

A descriptive schematic of the present study is presented in Figure 1**.** The SNPs were queried from the Taiwan Biobank dataset using an Extreme Gradient Boost (XGBoost) machine learning algorithm. SNPs connected to other SNPs in the network were retained. Next, we performed functional annotation of the SNPs according to the five aforementioned sets of functional annotations (missense, *cis*-eQTL, KEGG, PPI, and KO mice) using HaploReg V4.1. The prioritization of depression-associated genes was based on a scoring system comprising the five sets of functional annotations. The genes that were prioritized and identified as depression risk genes were converted and extended using the STRING database. Thereafter, overlapping of gene targets and drugs was identified using the DrugBank database. Finally, validation was performed using ClinicalTrials.gov and PubMed for drugs that were undergoing clinical trials and preclinical (in vitro and in vivo) studies, respectively.

### 2.2. Genes Associated with Depression

The SNPs identified using the machine learning algorithm were input into HaploReg v4.1 for functional annotation [62]. HaploReg v4.1 provides thorough information regarding genomic variants and changes in proteins by integrating various functional annotations [62]. Accordingly, the SNPs that encoded the genes for depression were obtained, and the list of the genes was used in subsequent analyses.

### 2.3. Five Sets of Functional Annotations for Prioritizing Genes Associated with Depression

A scoring system indicating the most promising target genes integrating the five sets of functional annotations was constructed. The sets of functional annotations were as follows: (i) Missense, to conduct missense functional annotations, we used RStudio v3.4.3 and the HaploR package [63], which contains annotations of functional consequences from a database of SNPs (dbSNPs). Because changes in the amino acid sequences might alter protein function, missense or nonsense variants can be considered as one of the important functional annotations. The genes with missense SNPs associated with depression were assigned 1 point. (ii) c*is*-eQTLs, a *cis*-eQTL SNP affects the expression of the gene at the location of the SNP [64]. The SNP is linked to a shift in gene expression in the target tissue, which has physiological consequences. Any gene with a *cis*-eQTL SNP associated with depression expressed in whole blood was given 1 point. (iii) KEGG, the KEGG, an online biochemical route database, was used to perform molecular pathway enrichment analysis [65]. Genes that were abundant in the KEGG pathway (false discovery rate (FDR) of 0.05) were each assigned 1 point [66]. (iv) PPI, the biological process category of gene ontology was used as a data source. An FDR of 0.05 was established as the threshold for significance [66]. (v) KO mice, to query the mouse phenotype, BioMart was used to convert the human gene ensemble IDs to mouse gene ensemble IDs [67]. The Mammalian Phenotype Ontology Browser, which includes information on mice and other mammalian phenotypes, was used as a data source. The gene set was considered significant when the FDR in the enrichment analysis was <0.05.

According to our functional annotations, genes with one functional annotation were assigned 1 point, and genes with a score of ≥2 points were identified as biological depression risk genes.

### 2.4. STRING and DrugBank Analysis

The STRING database provides information related to gene-encoded proteins. The identified depression risk genes were subjected to STRING analysis according to the proteins that they encoded [68]. The proteins encoded by the identified genes were considered potential drug targets, and were subjected to further analysis conducted using DrugBank, a large database (https://www.drugbank.ca/, accessed on 17 February 2022) with data for over 17,000 drug targets and 10,000 drug compounds [69].

### 2.5. Validation of Target Genes for Depression

The drugs identified from DrugBank were confirmed through two databases: ClincalTrial.gov (https://clinicaltrials.gov/, accessed on 19 February 2022) was used for the drugs undergoing human trials, and PubMed (https://pubmed.ncbi.nlm.nih.gov/, accessed on 19 February 2022) was used for the drugs undergoing preclinical (in vitro and in vivo) studies.

## 3. Results

We identified 5885 SNPs associated with depression (Appendix A), 632 of which were unique. The genes with the identified SNPs were identified as depression-associated genes (Appendix A).

### 3.1. Depression Risk Genes Identified Using Functional Annotations

We assigned each of the 632 unique depression-associated genes a score according to their functional annotations. The distribution of the functional annotations is illustrated in Figure 2. We used the missense variant and cis-eQTL annotations as the first and second criteria, respectively, for identifying and prioritizing the depression-associated genes. Overall, 34 and 68 of the depression-associated genes had missense and cis-eQTL SNPs, respectively. The third set of criteria for consideration, in terms of a depression-associated gene, were the gene ontology annotations. We identified 87 depression-associated genes. The fourth set of criteria were the PPI annotations. We identified 59 genes that overlapped with the depression-associated genes. The fifth set of criteria, the KEGG annotations, were used to perform an enrichment analysis on the molecular pathways. Sixteen depression-associated genes were identified in the KEGG-annotated pathways according to the enrichment analysis.

We compiled the scores of each of the genes (from 0 to 4 points) according to their functional annotations (Figure 3). The largest proportion of the genes (460 genes) had scores of 0 points. A total of 65 genes had scores ≥2, and were thus identified as depression risk genes (Table 1). Only seven of the genes—Interleukin 4 (*IL4*), Interleukin 18 Receptor 1 (*IL18R1*), Interleukin 6 Receptor (*IL6R*), Signal Transducer and Activator of Transcription 6 (*STAT6*), SMAD Family Member 3 (*SMAD3*), Interleukin 13 (*IL13*), and Toll-Like Receptor 1 (*TLR1*)—had scores of 4 points.

### 3.2. STRING Database for Gene Set Expansion

The STRING database, which combines publicly available data on direct (physical) and indirect (functional) protein–protein interactions, was used to extend the gene set of the 65 depression risk genes. Fifty interactions were selected from the database, and ultimately, 115 genes were selected as target genes and used in subsequent analyses (Appendix A).

### 3.3. Prioritization of Drug Repurposing Candidates for Depression

The DrugBank database was used to identify the druggable genes from among the 115 genes identified in the STRING analysis. Unfortunately, not all of the depression risk genes were druggable; only 19 of the genes were identified as druggable, and were determined as able to bind with 58 drugs. Of the seven genes with a score of 4 points, only *IL6R* was determined as druggable. All the identified target genes and drugs are listed in Appendix A.

Intriguingly, of the 58 identified drugs, 20 were undergoing clinical trials or preclinical studies for depression (Table 2). The other 38 drugs were new drugs that had never been previously reported as being used for the treatment of depression.

The target genes were those reported in preclinical studies and clinical trial studies to be the most promising target genes for depression. We identified nine target genes, including CD3 delta subunit of the T-cell receptor complex (*CD3D*), CD247 molecule (*CD247*), adenosine A1 receptor (*ADORA1*), cholinergic receptor nicotinic alpha 2 subunit (*CHRNA2*), protein kinase C epsilon (*PRKCE*), ferritin light chain (*FTL*), interleukin 5 (*IL5*), gamma aminobutyric acid type B receptor subunit 1 (*GABBR1*), and *IL6R.* Of the 38 new drugs, the following 15 targeted six of the most promising target genes: sodium ferric gluconate complex, ferric pyrophosphate citrate, blinatumomab, reslizumab, sarilumab, satralizumab, aminophylline, oxtriphylline, metocurine iodide, doxacurium, tubocurarine, decamethonium, metocurine, pancuronium, and pipecuronium (Table 3). Of these, we highlight sarilumab and satralizumab as exhibiting the most potential as drug repurposing candidates for depression because they target *IL6R*, which was identified as the gene exhibiting the strongest potential as a target gene according to the functional annotation scoring system and the validation conducted using the ClinicalTrials.gov and PubMed databases (Table 3).

## 4. Discussion

This study integrated machine learning and functional annotations to identify drug repurposing candidates for the treatment of depression. We identified seven key depression risk genes according to their highest functional annotation scores, and identified *IL6R* as the most promising target gene for depression according to clinical and preclinical evidence. In addition, we identified approximately 20 drugs undergoing clinical trials and preclinical studies for use in the treatment of depression, and 15 new drug repurposing candidates, including sarilumab and satralizumab, exhibiting strong potential for use in the treatment of depression. These findings indicate that adopting a genomic-analysis-based approach to drug repurposing can facilitate the discovery of new drugs for treating depression.

*IL6R* was one of the target genes with the highest functional annotation score and was a highly promising target in the treatment of depression. *IL6R* regulates systemic inflammation, which is associated with depression development [70,71,72]. Genetic variants of *IL6R* are associated with interleukin 6 (IL6) and C-reactive protein (CRP) regulation [73]. Furthermore, the upregulation and downregulation of IL6 and CRP affect depression severity [70,73]. Two major signaling pathways of IL6, classical signaling (anti-inflammatory) and trans-signaling (pro-inflammatory), were assumed to be related to depression development [74]. The classical signaling pathway occurs when IL6 binds with *IL6R* (a membrane-bound receptor) [75,76], while the trans-signaling pathway involves the attachment of IL6 to soluble interleukin 6 receptor (sIL6R), a non-membrane-bound receptor [76,77]. The activity of IL6 in the brain was often induced by trans-signaling [78,79,80,81]. A recent Mendelian randomization study showed that an increased number of sIL6Rs in the trans-signaling pathway significantly enhances the risk of depression [82]. In addition, high levels of sIL6R are associated with lower CRP production through classical signaling, which indicates a high risk of depression [82]. Tocilizumab, which is undergoing clinical studies under accession number NCT03787290, is a humanized monoclonal antibody that targets *IL6R*, thereby inhibiting IL6 classic signaling and trans-signaling pathways [83], and is effective at alleviating depressive symptoms [84]. An interventional study reported the benefit of tocilizumab in decreasing depression severity [85]. In addition, we identified two other drugs that target *IL6R*: sarilumab and satralizumab. Although no evidence regarding the use of these two drugs in the treatment of depression has yet been uncovered, they exhibit strong potential as drug repurposing candidates for depression.

*IL5* encodes a cytokine that is an effector cytokine of activated Th2 cells; that is, *IL5* activates Th cells after the cells are activated by IL4 [86]. Depression was associated with a lower IFN-γ level and an elevated IL13 level; the functions of IL13 have similarities with the role of *IL5* [87]. Thus, *IL5* might be associated with depression. This was supported by a gene set analysis study, in which *IL5* was upregulated in the post-mortem brain tissue of a patient with MDD [88]. A study that investigated the association between *IL5* and MDD in 116 participants (MDD = 58; control = 58) revealed that every 1-unit increase in serum *IL5* level is associated with a 76% greater risk of MDD [64]. In addition, a study by Tzang et al. observed that *IL5* level is associated with depression symptoms in cancer patients [89]. A possible mechanism has been proposed that increased amounts of cortisol in the circulation cause aberrant IL-5 cytokine production and secretion patterns, which in turn cause depressed symptoms [90]. Mepolizumab is a fully humanized recombinant IgG1 kappa monoclonal antibody against *IL5*, and has been approved for severe asthma [91]. In a previous study, mepolizumab administered for 6 months significantly reduced the occurrence of asthma exacerbations (from 48% to 38%) in patients with asthma and comorbid depression [92]. Mepolizumab is undergoing clinical trials for depression in patients with asthma (accession number: NCT-04680611). Another drug candidate identified in the present study is reslizumab, which targets *IL5*. We suggest that the mechanisms underlying the effect of reslizumab on the pathophysiology of depression involve *IL5*.

*CHRNA2* is a widely expressed subunit of nicotinic acetylcholine receptors, and is involved in neurocognitive disorders and nicotine dependence [93,94,95,96]. The position of *CHRNA2* in chromosomes (in the 8p region) may be involved in neurodegenerative and psychiatric disorders [96]. A study on prenatal depression patients found that differentially methylated *CHRNA2* related to antidepressant treatment [97]. In other words, *CHRNA2* was considered to be involved in depression pathogenesis. In the present study, carbamoylcholine, cisatracurium, atracurium besylate, mivacurium, vecuronium, and two drugs of which the clinical efficacy was confirmed through clinical trials (mecamylamine and ruconium) were determined to target *CHRNA2*. The non-competitive antagonist mecamylamine, a widely used therapeutic agent that targets acetylcholine receptors, may be effective in depression treatment [98]. In addition, reconium, originally used as a muscle relaxant, may have antidepressant effects, and is an effective adjunctive treatment with electroconvulsive therapy (ECT) [99,100]. Rocunium has also been observed to reduce myalgia and headache, and shorten the awakening time (spontaneous respiration and opening the eyes in response to verbal stimuli) after ECT [100].

Another target gene that we identified in the present study was *ADORA1*, which regulates various biological functions, including the mechanisms underlying sleep [101,102], and psychiatric disorders including depression [103,104,105]. *ADORA1* activation has been found to induce antidepressant-like effects [106,107]. In addition, the therapeutic effects of sleep deprivation [108] and ECT [109] are mediated by the activation or upregulation of *ADORA1*. Szopa et al. suggested a new approach in the treatment of people with depressive disorders that involves combining selective A1 and A2A receptor antagonists with magnesium or zinc [110]. Tramadol, a drug undergoing phase IV clinical trials for depression, was discovered to target *ADORA1* in the present study. Bumpus [111] assessed patients’ perceptions of the effectiveness and safety of tramadol as an off-label antidepressant relative to 34 other antidepressants, and discovered that most (94.6%) of the patients viewed tramadol as an effective antidepressant. Tramadol is a mu-receptor opioid agonist that increases the concentrations of serotonin and noradrenaline in the limbic system, thereby exerting an antidepressant effect [112]. In addition to tramadol, we identified other drugs linked to *ADORA1*, including caffeine, theophylline, adenosine, and pentoxifylline, that were undergoing phase 1 and 2 clinical trials. Furthermore, we discovered other target genes and drug repurposing candidates for depression, the efficacies of which are supported by published evidence; for example, muromanab, which targets *CD3D*/*CD247* [113,114], and taurine, which targets *GABBR1* [115,116].

In addition, in terms of neuroinflammation, *STAT6* was found to be associated with neurodegeneration diseases, including depression [117,118]. Interestingly, *STAT6* exhibited one of the highest scores based on the five functional annotations in the present study. Several previous studies support the role of *STAT6* in depression; these were validated in a preclinical investigation, in which *STAT6* signaling was discovered to be involved in some of the brain’s mechanisms, such as the activity of neurons and neuroplasticity [119,120]. Previous studies using animal models emphasized that a deficiency of *STAT6* decreases levels of dopamine and serotonin transporter; thus, *STAT6* is suggested to play a pivotal role in the pathogenesis of depression through monoamine regulation in the hippocampus of the brain [119,121]. To date, this result has not been confirmed in clinical studies. Unfortunately, the drug target genes that we identified are not all involved in pharmacological activities (undruggable), including STAT6. However, we propose that STAT6 can be considered as a potential biomarker for depression.

Drug repurposing offers several advantages, such as a shorter time period, being cost effective, and imposing less risk compared to traditional drug discovery [122]. Additionally, in this study, drug repurposing by genomic analysis presents the strength of its ability to bridge the gap between genomic medicine and conventional personalized trials for the treatment of depression by offering new perspectives on pharmacogenomic-guided medication based on biological depression risk genes of depression patients. Despite the fact that our study demonstrates the feasibility and value of using SNP data to determine drug repurposing candidates for the treatment of depression, it still has some limitations. Not all SNPs are biologically significant, and not all the identified depression risk genes could be targeted by drugs. Moreover, due to the nature of present study, we were unable to investigate the therapeutic effects of our findings. For future research, more functional in vitro and in vivo investigations (in primary basic/preclinical research or clinical trials) and validations are required.

## 5. Conclusions

In this study, we reveal a schematic approach for the use of functional genomic variations to identify drug target genes and potential repurposed drugs prior to preclinical and clinical trial studies for depression. Our findings propose *IL6R* as the most promising target gene for depression due to *IL6R* exhibiting the highest functional annotation score and its validation in ClinicalTrial.gov and PubMed databases. In addition, we identified two candidate drugs (sarilumab and satralizumab) with strong potential use in the treatment of depression. In summary, this study indicates that using a genomic-analysis-based approach to discover drugs for treating depression is both time- and cost-effective. Furthermore, the findings of our study can serve as evidence of genes related to the inflammatory pathway, and provide new insight into pathomechanism for depression. Future studies need to investigate the role of *IL6R* in the pathogenesis of depression, as well as the interactions between *IL6R* and sarilumab or satralizumab.

## Figures and Tables

**Figure 1 biomedicines-10-01947-f001:**
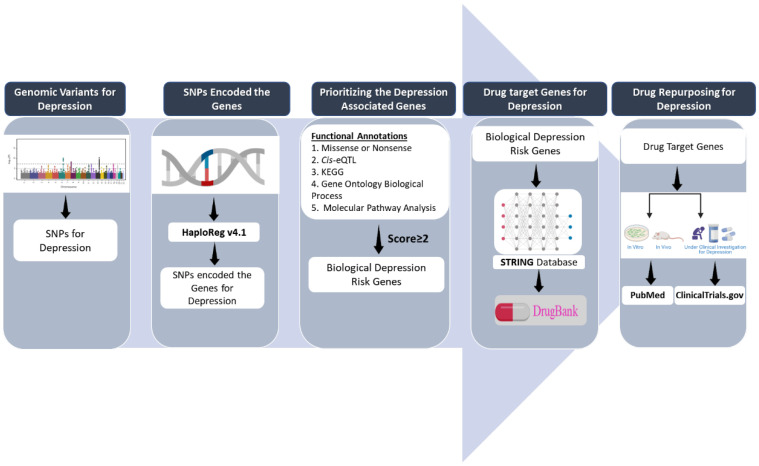
Overview of drug repurposing for depression. In this study design, SNPs were prioritized using a machine learning algorithm and various databases: HaploReg v4.1, STRING, DrugBank, ClinicalTrials.gov, and PubMed.

**Figure 2 biomedicines-10-01947-f002:**
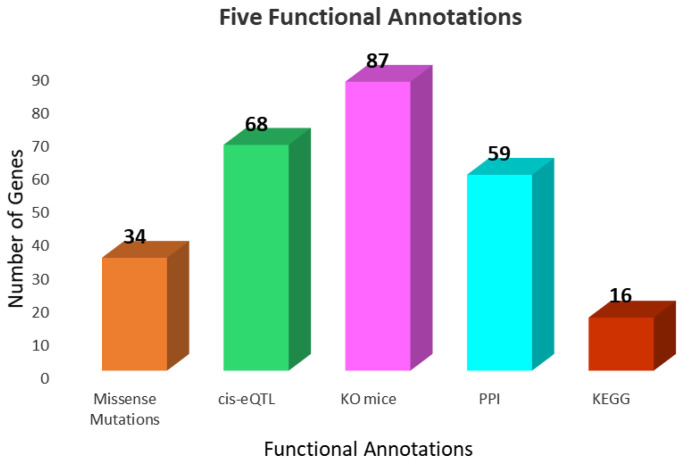
Histogram distribution of functional annotations.

**Figure 3 biomedicines-10-01947-f003:**
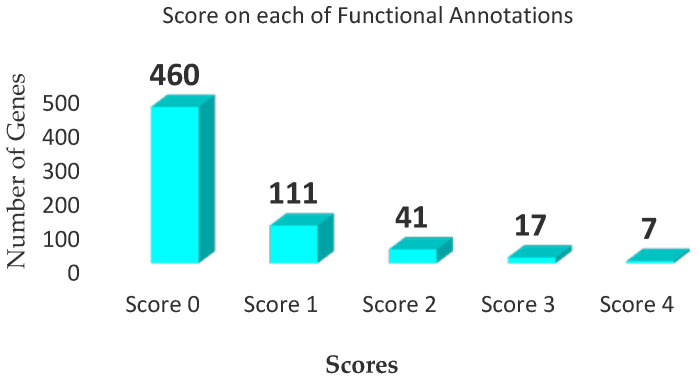
Histogram distribution of gene scores: 460 and 111 genes had scores of 0 and 1, respectively; the 65 genes with total scores ≥2 were identified as “depression risk genes”.

**Table 1 biomedicines-10-01947-t001:** Five functional annotations applied to prioritize the depression risk genes.

GENCODE_ID	GENCODE_Name	Missense Variant	Cis-eQTL	KEGG	PPI	KO Mice	Total Score
ENSG00000113520	*IL4*	0	1	1	1	1	4
ENSG00000115604	*IL18R1*	0	1	1	1	1	4
ENSG00000160712	*IL6R*	1	1	0	1	1	4
ENSG00000166888	*STAT6*	0	1	1	1	1	4
ENS G00000166949	*SMAD3*	0	1	1	1	1	4
ENSG00000169194	*IL13*	1	0	1	1	1	4
ENSG00000174125	*TLR1*	1	1	0	1	1	4
ENSG00000020633	*RUNX3*	0	1	0	1	1	3
ENSG00000069667	*RORA*	0	0	1	1	1	3
ENSG00000107485	*GATA3*	0	0	1	1	1	3
ENSG00000109471	*IL2*	0	0	1	1	1	3
ENSG00000113525	*IL5*	0	0	1	1	1	3
ENSG00000115602	*IL1RL1*	1	0	0	1	1	3
ENSG00000117586	*TNFSF4*	0	1	0	1	1	3
ENSG00000125347	*IRF1*	0	1	0	1	1	3
ENSG00000134215	*VAV3*	0	1	0	1	1	3
ENSG00000138684	*IL21*	0	0	1	1	1	3
ENSG00000141736	*ERBB2*	1	0	0	1	1	3
ENSG00000158869	*FCER1G*	0	1	0	1	1	3
ENSG00000161405	*IKZF3*	0	1	0	1	1	3
ENSG00000179344	*HLA-DQB1*	0	1	1	0	1	3
ENSG00000204252	*HLA-DOA*	0	0	1	1	1	3
ENSG00000204287	*HLA-DRA*	1	1	1	0	0	3
ENSG00000231389	*HLA-DPA1*	0	1	1	1	0	3
ENSG00000073605	*GSDMB*	1	1	0	0	0	2
ENSG00000074047	*GLI2*	0	0	0	1	1	2
ENSG00000079112	*CDH17*	0	0	0	1	1	2
ENSG00000087086	*FTL*	0	0	0	1	0	2
ENSG00000087088	*BAX*	0	0	0	1	1	2
ENSG00000100385	*IL2RB*	0	1	0	0	1	2
ENSG00000100902	*PSMA6*	0	1	0	0	0	2
ENSG00000106571	*GLI3*	0	0	0	1	1	2
ENSG00000107957	*SH3PXD2A*	1	0	0	0	1	2
ENSG00000111145	*ELK3*	1	0	0		1	2
ENSG00000111335	*OAS2*	1	1	0		0	2
ENSG00000112130	*RNF8*	0	0	0	1	1	2
ENSG00000112486	*CCR6*	0	0	0	1	1	2
ENSG00000113522	*RAD50*	0	0	0	0	1	2
ENSG00000120903	*CHRNA2*	0	1	0	0	1	2
ENSG00000124107	*SLPI*	0	0	0	1	1	2
ENSG00000131507	*NDFIP1*	0	0	0	1	1	2
ENSG00000134460	*IL2RA*	0	0	0	1	1	2
ENSG00000134470	*IL15RA*	0	1	0	0	1	2
ENSG00000135905	*DOCK10*	0	0	0	1	1	2
ENSG00000137033	*IL33*	0	0	0	1	1	2
ENSG00000142556	*ZNF614*	1	1	0	0	0	2
ENSG00000143631	*FLG*	1	0	0	0	1	2
ENSG00000145777	*TSLP*	0	0	0	1	1	2
ENSG00000162104	*ADCY9*	0	1	0	0	1	2
ENSG00000163485	*ADORA1*	0	1	0	0	1	2
ENSG00000165280	*VCP*	0	0	0	1	1	2
ENSG00000167914	*GSDMA*	1	1	0	0	0	2
ENSG00000171132	*PRKCE*	0	0	0	1	1	2
ENSG00000171608	*PIK3CD*	0	0	0	1	1	2
ENSG00000172057	*ORMDL3*	0	1	0	1	0	2
ENSG00000174130	*TLR6*	0	0	0	1	1	2
ENSG00000179588	*ZFPM1*	0	0	0	1	1	2
ENSG00000180902	*D2HGDH*	1	1	0	0	0	2
ENSG00000186265	*BTLA*	0	0	0	1	1	2
ENSG00000186716	*BCR*	0	1	0	1	0	2
ENSG00000196735	*HLA-DQA1*	0	1	1	0	0	2
ENSG00000197746	*PSAP*	0	0	0	1	1	2
ENSG00000198821	*CD247*	0	1	0	0	1	2
ENSG00000204681	*GABBR1*	1	0	0	0	1	2
ENSG00000215182	*MUC5AC*	1	0	0	0	1	2

Note: The italic font in table indicates genes name; the colors of missense variant: orange, cis-eQTL: green, KEGG: red, PPI: blue, KO mice: purple meaning scored as 1; The darker of grey color in total score meaning higher scores.

**Table 2 biomedicines-10-01947-t002:** Pharmacological therapies in development for the treatment of depression.

Gene	Drug	Original Indication	Identifier * (NCT-0/PMID)
**ClinicalTrials.gov**			
*FTL*	Iron Dextran	Iron deficiency	3373253
*IL5*	Mepolizumab	Eosinophilic granulomatosis with polyangiitis (EGPA)	4680611
*IL6R*	Tocilizumab	Rheumatoid arthritis	3787290
*ADORA1*	Tramadol	Moderate to severe pain	3309163
*ADORA1*	Caffeine	Migraine	0025792
*ADORA1*	Theophylline	Chronic asthma	1263106
*ADORA1*	Adenosine	Tachycardia	2902601
*ADORA1*	Pentoxifylline	Intermittent claudication	4417049
*PRKCE*	Tamoxifen	Breast cancer	0667121
*CHRNA2*	Mecamylamine	Hypertension	0593879
*CHRNA2*	Rocuronium	General anesthesia	4565730
*GABBR1*	Taurine	Total parenteral nutrition	0217165
**PubMed**			
*CD3D*	Muromonab	Prevention of organ rejection	24257035
*CD247*	Muromonab	Prevention of organ rejection	24257035
*ADORA1*	Dyphylline	Asthma	10064181
*CHRNA2*	Carbamoylcholine	Open-angle glaucoma	23603524
*CHRNA2*	Cisatracurium	General anesthesia	22092267
*CHRNA2*	Atracurium besylate	General anesthesia	8442962
*CHRNA2*	Mivacurium	General anesthesia	8346843
*CHRNA2*	Vecuronium	Muscle relaxant	8733812

Note: * Identifiers from ClinicalTrials.gov and PubMed database; the italic font indicates genes name.

**Table 3 biomedicines-10-01947-t003:** Drug repurposing candidates for depression identified using a genomic-analysis-based approach.

Biological Gene	Target Drug	Original Indication	Score
*IL6R*	Sarilumab	Rheumatoid arthritis	4
*IL6R*	Satralizumab	Neuromyelitis optica spectrum disorder (NMOSD)	4
*IL5*	Reslizumab	Severe asthma	3
*sFTL*	Sodium ferric gluconate complex	Iron deficiency anemia	2
*FTL*	Ferric pyrophosphate citrate	Iron deficiency	2
*CD3D*	Blinatumomab	Acute lymphoblastic leukemia (ALL)	2
*ADORA1*	Aminophylline	Asthma	2
*ADORA1*	Oxtriphylline	Asthma	2
*CHRNA2*	Metocurine iodide	Muscle contractions	2
*CHRNA2*	Doxacurium	General anesthesia	2
*CHRNA2*	Tubocurarine	General anesthesia	2
*CHRNA2*	Decamethonium	Muscle relaxant	2
*CHRNA2*	Metocurine	Muscle relaxant	2
*CHRNA2*	Pancuronium	Muscle relaxant	2
*CHRNA2*	Pipecuronium	Muscle relaxant	2

Note: Scores were obtained from a scoring system based on five sets of functional annotations; the italic font indicates genes name.

## Data Availability

The data presented in this study are available on request from the corresponding author. The data are not publicly available due to privacy.

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
