# Peer review of "Genomic-Analysis-Oriented Drug Repurposing in the Search for Novel Antidepressants"

_biomedicines, 2022, doi:10.3390/biomedicines10081947_

Round 1

Reviewer 1 Report

The manuscript biomedicines-1802071 entitled Drug Repurposing for Depression Based on Genomic Analysis by Moh. Hendra Setia Lesmana and co-workers, presented a study to identify drug-repurposing candidates for the treatment of depression by adopting a genomic analysis–based approach. Their findings indicate that a genomic analysis–based approach can facilitate the discovery of drugs that can be repurposed for treating depression.

The scientific work was well conducted: the experimental work is consistent with hypothesis and the methodology adequate.

The discussion is consistent with results.

Very interesting the importance of IL6, PCR and IL5 with depression: it was already well described a connection between inflammation and depression.

Also they authors described in addiction to IL6 other target genes as CHRNA2, ADORA1 and GABBR1.

The tables and figures should be better formatted for the manuscript. Some tables are divided in two pages (see table 2 and 3 for example).

There is a section 0 at line 45 which should be removed.

A linguistic and stylistic revision is recommended.

Author Response

Reviewer 1.

The manuscript biomedicines-1802071 entitled Drug Repurposing for Depression Based on Genomic Analysis by Moh. Hendra Setia Lesmana and co-workers, presented a study to identify drug-repurposing candidates for the treatment of depression by adopting a genomic analysis–based approach. Their findings indicate that a genomic analysis–based approach can facilitate the discovery of drugs that can be repurposed for treating depression.

Comments and Suggestions for Authors:

Q1: Reviewer #1. The scientific work was well conducted: the experimental work is consistent with hypothesis and the methodology adequate.

A1: We thank the reviewer’s comments

Q2: Reviewer #1. The discussion is consistent with results.

A2: We sincerely thank the reviewer’s comment.

Q3: Reviewer #1. Very interesting the importance of IL6, PCR and IL5 with depression: it was already well described a connection between inflammation and depression.

A3: We really appreciate the reviewer’s comment.

Q4: Reviewer #1.  Also, the authors described in addition to IL6 other target genes as CHRNA2, ADORA1 and GABBR1.

A4: We really appreciate the reviewer’s comment.

Q5: Reviewer #1. The tables and figures should be better formatted for the manuscript. Some tables are divided in two pages (see table 2 and 3 for example).

A5: We thank to the reviewer’s suggestion. We replace the figure 1 to be better formatted according to the reviewer’s concern (Page 3). We also reformatted the tables to be more easily read, such as (Table 1 in page 6 and 7; Table 2 in page 8; Table 3 in page 9) according to reviewer’s concern.

Q6: Reviewer #1. There is a section 0 at line 45 which should be removed.

A6: We are very grateful to the reviewer’s suggestion. We removed already section 0 from line 45-51. Thank you

Q7: Reviewer #1. A linguistic and stylistic revision is recommended.

A7: We really appreciate the reviewer’s suggestion. We invited our native English college that is Dr. Ismaila Sonko to check and revise the grammar of manuscript to be properly written according to reviewer’s concern. Thank you

Reviewer 2 Report

In the present study Dr. Lesmana and colleagues evaluated drug-repurposing candidates for the treatment of depression adopting a genomic analysis–based approach.

The manuscript is interesting, however the authors may need to clarify a couple of points.

Comments:

-       The association between depression and interleukin-6 signaling may need to be further discussed. In this regard, the current literature may need to be updated (e.g  Brain Behav Immun. 2021 Jul;95:106-114.  doi: 10.1016/j.bbi.2021.02.019).

-       The authors found an interesting association between STAT6 gene and depression. However, this topic was not discussed. The authors may need to discuss the importance of STAT6 in depression in both pre-clinical and clinical studies. Is STAT6 a suitable pharmacological target? Moreover, STAT6 has also been correlated with inflammation and cytokine production. Please, discuss this critical point.

Author Response

Reviewer 2.

In the present study Dr. Lesmana and colleagues evaluated drug-repurposing candidates for the treatment of depression adopting a genomic analysis–based approach.

The manuscript is interesting; however, the authors may need to clarify a couple of points.

Comments:

Q1: Reviewer #2. The association between depression and interleukin-6 signaling may need to be further discussed. In this regard, the current literature may need to be updated (e.g  Brain Behav Immun. 2021 Jul;95:106-114.  doi: 10.1016/j.bbi.2021.02.019).

A1: We are very grateful to the reviewer’s suggestions. We did it as suggested by the reviewer. We already adjusted the additional explanation sentences in the discussion part as suggested by the reviewer. The additional of sentences are presented in the following paragraph [Page 9, lines 242-245]

Q2: Reviewer #2.  The authors found an interesting association between STAT6 gene and depression. However, this topic was not discussed. The authors may need to discuss the importance of STAT6 in depression in both pre-clinical and clinical studies. Is STAT6 a suitable pharmacological target? Moreover, STAT6 has also been correlated with inflammation and cytokine production. Please, discuss this critical point.

A2: Many thanks to the reviewer’s suggestions. Yes, it is a very important point.  STAT6 was identified as one of the highest scores based on five-functional annotations in present study. Several studies supported the role of STAT6 directly with depression which were validated in preclinical investigation, STAT6 signalling was described to involve in some brain’s mechanisms, such as activity of neuron and neuroplasticity[50, 51]. Previous study in animal model emphasized that deficiency of STAT6 decreased level of dopamine and serotonin transporter, thus, STAT6 suggested play pivotal role in pathogenesis of depression through monoamines regulation in hippocampus of brain[50, 52]. To date, this result has not been confirmed in clinical study. Unfortunately, the drug target gene that we identified are not all in pharmacological activities (undruggable) including STAT6. However, we proposed that STAT6 can be considered as biomarker for depression. We already add the explanation of STAT6 gene and depression according to the reviewer’s suggestions. [Page 10-11, lines 301-312]
